# nc886, an RNA Polymerase III-Transcribed Noncoding RNA Whose Expression Is Dynamic and Regulated by Intriguing Mechanisms

**DOI:** 10.3390/ijms24108533

**Published:** 2023-05-10

**Authors:** Yeon-Su Lee, Yong Sun Lee

**Affiliations:** 1Rare Cancer Branch, Research Institute, National Cancer Center, Goyang 10408, Republic of Korea; 2Department of Cancer Biomedical Science, Graduate School of Cancer Science and Policy, National Cancer Center, Goyang 10408, Republic of Korea

**Keywords:** nc886, RNA polymerase III, regulation, transcription factor, CpG methylation

## Abstract

nc886 is a medium-sized non-coding RNA that is transcribed by RNA polymerase III (Pol III) and plays diverse roles in tumorigenesis, innate immunity, and other cellular processes. Although Pol III-transcribed ncRNAs were previously thought to be expressed constitutively, this concept is evolving, and nc886 is the most notable example. The transcription of nc886 in a cell, as well as in human individuals, is controlled by multiple mechanisms, including its promoter CpG DNA methylation and transcription factor activity. Additionally, the RNA instability of nc886 contributes to its highly variable steady-state expression levels in a given situation. This comprehensive review discusses nc886’s variable expression in physiological and pathological conditions and critically examines the regulatory factors that determine its expression levels.

## 1. Introduction

### 1.1. nc886 Is Exceptional among RNA Polymerase III (Pol III)-Transcribed Genes

Over the past two decades, there has been extensive research on non-coding RNAs (ncRNAs), particularly those with regulatory functions. This field was sparked by the discovery of microRNAs (miRNAs) and RNA interference and has been fueled by the development of high-throughput RNA-sequencing technology (RNA-seq), which captured cellular transcripts that are miniscule in quantity and so had been previously undetectable. Most of these transcripts lack an open reading frame (ORF) and are classified as ncRNAs. Currently, the number of ncRNAs in public databases is comparable to or even greater than that of protein-coding genes [1,2].

Despite the extensive research on ncRNAs, there is a class of ncRNAs that has been relatively neglected. This class includes ncRNAs that are approximately 70 to 200 nucleotides (nts) long and are transcribed by Pol III. Representative examples are transfer RNAs (tRNAs), 5S ribosomal RNA (5S rRNA), and U6 small nuclear RNA (U6 snRNA). These classic ncRNAs play fundamental roles in cellular metabolism, and their variable expression is difficult to comprehend. Therefore, it has been generally believed that Pol III-transcribed genes (shortly “Pol III genes”) are expressed constitutively. In fact, the expression level of a mature tRNA is high and mostly constant due to extensive nucleotide modifications and the secondary structure. Similarly, 5S rRNA and U6 snRNA are copious and expressed almost constantly. For researchers studying Pol III genes, there are practical hurdles when it comes to investigating the expression of Pol III genes. In many cases, identical or highly similar sequences of a Pol III gene are scattered at multiple loci in the genome. Thus, even when the altered expression of a Pol III gene is recognized, it is challenging to identify which locus accounts for the change. For these reasons, most researchers have overlooked the regulation of Pol III genes. Even when a researcher was interested, there exists a limitation to investigating Pol III regulation in-depth.

However, the perception of constant expression for classic Pol III genes has changed in recent times, with the recognition of diversity in the Pol III transcriptome and the dysregulation of Pol III genes in various biological conditions (reviewed in [3,4]). One of the most prominent examples of this is nc886, the topic of this review. nc886 is transcribed by Pol III but differs from classic Pol III genes in several ways. The nc886 gene exists at a single locus in the human genome and expresses highly unstable RNA that plays gene-regulatory roles (Figure 1A–D). Regarding dynamic expression and exquisite regulation, nc886 is likely more interesting than most protein-coding genes and ncRNAs that are transcribed by RNA polymerase II (Pol II).

### 1.2. nc886’s Identity and Function

nc886 is a 101- (or 102-) nts-long ncRNA (Figure 1) with multiple names, including vtRNA2-1, pre-miR-886, miR-886-5p, -3p, etc. Despite these names suggesting that nc886 may be a vault RNA (vtRNA) or a microRNA (miRNA), experimental data have shown that it is not either of these. Whereas a canonical vtRNA, vtRNA1-1, is a component of a nucleoprotein complex called the vault complex, nc886 does not exist in this complex [5,6]. Additionally, nc886 does not produce significant amounts of mature miRNAs [5,7].

Based on reports to date, nc886’s function is to bind target proteins, control their activity, and thereby regulate gene expression (Figure 1C). The direct interactors of nc886 that have been experimentally validated are Protein Kinase R (PKR), 2′–5′-oligoadenylate synthetase 1 (OAS), and Dicer [5,7,8,9,10]. In the case of Dicer, we should mention that its interaction with nc886 does not yield mature miR-886-5p and -3p [7], although it is a miRNA biogenesis enzyme. The roles of PKR, OAS, and Dicer have been well established in innate immunity and cancer. By controlling the activities of these proteins, nc886 affects several pathways, including the Nuclear Factor kappa-light-chain-enhancer of the activated B cells (NF-κB) pathway and the miRNA pathway. Several reports have indicated that nc886 is a determinant for apoptosis, tumorigenesis, viral replication, and immune responses [7,11,12,13,14,15,16]. Therefore, understanding how nc886 expression is controlled in these biological conditions is important.

## 2. Regulation of nc886 Expression

The expression of nc886 is controlled at multiple levels. The transcriptional rate is determined by transcription factors (TFs) and via CpG DNA methylation at its promoter region. Moreover, nc886 RNA is highly unstable, resulting in rapid changes in its intracellular levels in response to signals. In this review, we have compiled almost all of the literature on the regulation of nc886 expression. We will introduce the basic features of the nc886 gene, summarize biological situations in which its expression varies, and describe external and biological factors that control nc886 expression.

### 2.1. Pol III Promoter Elements of nc886

nc886 is undoubtedly transcribed by Pol III, as shown by its sequence signature and experimental evidence. The nc886 gene contains well-conserved internal promoter A and B boxes as well as consecutive thymidylates (TTTT) at the 3′-end for transcription termination (red letters in the nc886 sequence, Figure 1B). In addition, Chromatin ImmunoPrecipitation followed by high-throughput sequencing (ChIP-seq) clearly shows the occupation of Pol III enzyme subunits at the nc886 genomic region [17,18,19,20]. nc886 is not transcribed by Pol II because its expression was unaffected when a bladder cancer cell line was treated with a Pol II inhibitor, α-amanitin [21].

According to cis-elements and initiation factors to recruit Pol III catalytic units, Pol III promoters are categorized into three subtypes: type 1 for 5S rRNA, type 2 for tRNAs, and type 3 for U6 snRNA (reviewed in [22]). The elements for the type 1 and 2 promoters are located within the transcribed region, whereas the type 3 promoter is at the 5′-upstream of a transcription start site. ChIP-seq data and sequence analysis suggest that nc886 is most likely a type 2 promoter [20].

In type 2 promoters, A and B boxes are critical sequence elements, and this is also true for nc886. When its A and B boxes became deviant from the consensus sequence by mutagenesis, nc886 expression was significantly impaired [20]. A and B boxes are often referred to as gene-internal promoters, and since the definition of a promoter is a cis-element that drives transcription, these elements by themselves are supposed to do so. However, they are not sufficient, and 5′-upstream sequences are also required for nc886 transcription (Lee laboratory, unpublished data). This notion was supported by experimental data that showed that nucleosome depletion at the nc886 DNA region was associated with active transcription, and the depletion footprint was detected the 5′-upstream region [23]. Such an extra requirement of 5′-upstream sequence elements is also reported in some other Pol III genes such as 7SL RNA, a subset of tRNA genes, and an Epstein–Barr-virus (EBV)-encoded small RNA gene (EBER2) (reviewed in [24]). One study reported that the rat vtRNA gene has both type 2 and 3 promoters and that its transcription needs both internal and 5′-upstream elements [25]. However, the existence of a type 3 promoter element at the 5′-upstream of human nc886 is unlikely because ChIP-seq data failed to detect BRF2, an initiation factor specific for type 3 [17,18].

### 2.2. Measurement of nc886 Expression Levels

Before summarizing cases of variable nc886 expression (to be elaborated in the next section), we should mention the methods used to measure its expression levels. Generally, gene expression levels are measured using Northern blot or standard quantitative reverse transcription PCR (qRT-PCR) with two specific primers. These methods can be used for nc886, as it is an abundant 101 nt-sized RNA (Figure 1B). Northern hybridization is often used to visualize the intact 101 nt band and is considered the most reliable method for measuring nc886 expression levels [26].

It should be noted that nc886 was initially identified as a potential miRNA precursor and two mature miRNAs, miR-886-5p and miR-886-3p (Figure 1B), were included in the miRNA database (miRBase; https://www.mirbase.org/) for some time (miRBase version 10 to 15). However, it is now known that nc886 barely produces mature miRNAs, and functional studies on miR-886-5p and -3p should be interpreted with caution [26]. Nonetheless, the misidentification and inclusion of miR-886-5p and -3p in several miRNA array platforms ironically provided useful information about nc886 expression. Since miR-886-5p and -3p correspond to nc886’s 5′- and 3′-ends, respectively (Figure 1B), nc886 can be sensed by probes or primers that were originally designed for mature miRNAs. Actually, probes against miR-886-5p or -3p in hybridization-based arrays are nearly identical to those for conventional nc886 Northern. In PCR-based arrays, primers for miR-886-3p will amplify nc886 with comparable efficiency because they share almost the same 3′-ends. Our Northern blot experiments never observed smaller products than nc886 at a 101 nts size, and therefore, we are confident that it was intact nc886 but neither miR-886-5p/-3p nor degradation products, which was detected in most miRNA array experiments [5,7,26]. Therefore, we regarded a miRNA array value as an nc886 expression level and included the relevant literature in this review.

### 2.3. nc886 Expression Levels Are Variable in Diverse Biological Situations

As could be inferred by Pol III transcription, nc886 is abundantly present in most normal tissues. Actually, the expression was readily detectable by Northern hybridization without PCR amplification in normal tissues, including brain, breast, colon, liver, lung, and skeletal muscle tissues [5,20]. The intracellular copy number of nc886, which was calculated in HeLa cells, is 10^5^ RNA molecules per cell [5]. Normal tissues are estimated to have approximately 10^4^ copies of nc886 per cell [20]. These numbers indicate that the abundance of nc886 is comparable to that of small nucleolar RNAs (snoRNAs) or snRNAs and is at least 10-fold higher than most miRNAs.

nc886’s variable expression has been abundantly observed in the literature. A significant fraction of studies performed miRNA-profiling experiments and found that miR-886-5p or -3p were some of the top altered miRNAs. As stated earlier, they can be a proxy measure of nc886. We have summarized cohort studies in pathological conditions (Table 1 and Table 2), which include various types of cancer, immune-related diseases, and neurological disorders. In addition, a number of studies that presented cell culture and tissue data demonstrated that nc886 expression is altered by diverse biological and environmental factors (summarized in Table 3).

Tumorigenesis is the biological situation where nc886’s variable expression has been reported the most (Table 1). nc886 expression is increased in some tumor specimens but is decreased in others. The underlying mechanisms for these two opposite patterns are TF and epigenetic regulation, which were well investigated in human mammary epithelial cells (HMEC), a cell culture model system that recapitulates early breast tumorigenesis [20] (Figure 2). In this system, there are several lineages of immortalized and malignant cell lines serially derived from normal breast tissues that were obtained through reduction mammoplasty. When primary cells survive through the “stasis” cell death barrier and then become malignant via immortalization, nc886 gradually increases by MYC in one lineage but is silenced in another lineage via CpG DNA hypermethylation (Figure 2). nc886’s gradual increase during tumor progression was also shown in endometrial cancer and cervical cancer tissues [80,81]. In these two studies, the authors used fluorescence in situ hybridization (FISH) and observed that nc886 expression is low in normal tissues, medium in tumors at early stage, and high in advanced tumors. The rise of nc886 in these reports is in line with stronger Pol III activity in cancer [82]. On the other hand, nc886 appeared to decrease when an immortalized lung cell line BEAS2B was transformed to be malignant by treating with carcinogenic gas radon since miR-886-5p and -3p were within the top ten downregulated miRNAs [83].

Besides pathological states such as cancer, nc886 levels also change when cells differentiate, become activated, respond to a growth factor, or die. In addition, chemical compounds and viral infections affect nc886 expression. The hitherto reported cases are summarized in Table 3 and illustrated in Figure 3.

Several studies measured nc886 levels during T-cell differentiation/activation, after viral infection, and upon treatment with immunosuppressants (Table 3 and references therein). nc886 expression in these conditions deserves further discussion, given its role in innate immunity. nc886 is a suppressor of an anti-viral protein PKR [5,8,9] (Figure 1C). By controlling PKR and others, nc886 regulates pro-inflammatory NF-κB and the production of cytokines such as interferon-β (IFN-β), IFN-γ, interleukin-2 (IL2) [5,15,16,32,40,41]. Although nc886 is generally thought to suppress PKR and innate immunity, there are reports that a conformer of nc886 can activate PKR, and nc886 is required for cytokine production [9,16]. In this regard, we can interpret the significance of increased nc886 upon T-cell activation in two ways. On one hand, nc886 might act in favor of cytokine production. On the other hand, nc886 might prevent T-cells from being hyper-activated. nc886 expression becomes high upon infection of some viruses (EBV, Kaposi’s sarcoma virus, influenza A virus) [14,78] but low by others (cytomegalovirus, parvovirus, AdV) [13,78] (Figure 3 and Table 3). These results are intriguing and raise several questions. Which factor(s) determines increase/decrease of nc886 in individual cases of viral infection? Is it a host factor or a viral factor? Does nc886 act in a pro-viral way or anti-viral way? The host–virus interaction is complicated even within a viral infection cycle. For example, nc886 expression decreases when AdV replicates, although nc886 is required for AdV replication [13]. Extensive research efforts and the accumulation of data will be needed to delineate altered nc886 expression and its biological consequences during viral infection since nc886 appears to respond and act differently among viruses, host cell types, and even time points during a viral life cycle.

nc886 expression during cellular senescence was reported in two studies: one in primary human umbilical vein endothelial cells (HUVECs) [71] and the other in human dermal fibroblasts (HDF) [72]. They yielded conflicting data. (Figure 3 and Table 3). Notably, nc886’s impact on cell proliferation or death is two-faced. nc886 knockdown (KD) provokes the PKR cell death pathway, leading to apoptosis [5,6,7,12,32,40,41]. Ectopically expressed nc886 is anti-proliferative by delaying cell cycle progression [40,41,84]. It remains to be demonstrated whether these two different ways for nc886 to impair cell proliferation could explain the two opposite results in HUVEC and HDF during cellular senescence.

nc886 expression also changes in response to various external triggers, including chemotherapeutic drugs, natural products, and ultraviolet light (Figure 3 and Table 3). The most remarkable ones are doxorubicin and aclarubicin, which are anthracycline family compounds. They inhibit DNA metabolism, are expected to be specific to dividing cells, and so are used in chemotherapy for cancer patients. Doxorubicin suppresses nc886 expression by instantly evicting the Pol III enzyme machinery from DNA [6]. Although this mechanism globally inhibits Pol III transcription, nc886 decreases much earlier than most other Pol III-transcribed RNAs owing to its short half-life [5,20]. As aforementioned, nc886 depletion leads to PKR activation and consequently apoptosis. These events precede DNA-damage-induced cell death because nc886 level drops so quickly upon doxorubicin treatment. Importantly, this provides a reason why some chemotherapeutic drugs are cytotoxic not only to cancer cells but also to non-dividing normal cells in which the nc886-PKR cell death pathway operates. The doxorubicin/nc886 case explicitly shows the clinical utility of understanding the mechanisms responsible for how nc886 expression is controlled. When a chemical compound or a natural product is used for therapeutic or dietary purposes, nc886 expression should be taken into account when predicting efficacy and potential side effects.

nc886 expression levels appear to differ among healthy persons, as suggested by a number of epigenetic cohort studies, which are beyond the scope of this review and will not be discussed here. However, we should note that at least two cohort studies have measured the levels of nc886 RNA (or miR-886-5p and -3p as a proxy) in blood samples of healthy individuals, demonstrating varying inter-individual expression levels. These cohorts include the Young Finns study (YFS) cohort (*n* = 47) [67] and a group of women who delivered term birth (*n* = 20) as a comparison group for pre-term birth [66].

### 2.4. Factors to Determine nc886 Expression; TFs

TFs that are well-known in cancer, such as MYC and p53, are presumed to affect nc886 transcription, because they bind to the Pol III enzyme machinery to control its activity (reviewed in [82]). In addition, ChIP-seq data from the Encyclopedia of DNA Elements (ENCODE) project suggest the occupation of a number of TFs in the nc886 genomic region. We conjecture that those TFs control nc886 expression during pathological conditions, cellular stresses, and other situations (Table 1, Table 2 and Table 3). However, there is limited experimental data available on only a few TFs, including MYC and E2F1.

The role of MYC in nc886 expression was clearly demonstrated in the HMEC breast tumorigenesis model system [20]. In this system, nc886 expression increased when normal mammary cells become malignant, and this increase was mostly attributed to MYC (Figure 2). MYC bonded to the nc886 promoter region regardless of an E-box, the consensus DNA sequence that MYC recognizes. Although there are three E boxes at 1–2 kb upstream of nc886, they are unoccupied by MYC and dispensable to nc886 transcription. Most probably, MYC directly associates with the Pol III enzyme machinery and stimulates its binding to the nc886 promoter because the ChIP footprint of MYC coincided with that of a Pol III catalytic subunit POLR3A, and direct interaction between MYC and Pol III machinery was previously documented [85].

Another study reported that E2F1 activates nc886 expression [81]. In cervical cancer cell lines, overexpression of E2F1 resulted in an increase in nc886 expression, whereas treatment with caffeine, an E2F1 inhibitor, led to a decrease in nc886 expression. The involvement of E2F1 was further investigated using luciferase assays, where various DNA segments of the nc886 promoter region were cloned into a luciferase vector. This study claimed that the activation of the luciferase activity was dependent on the presence of predicted E2F binding sites. However, it should be noted that this study employed the pGL3 promoter vector, and so, the luciferase expression was from the SV40 promoter, which is a Pol II promoter. This is incompatible with nc886, which is transcribed by Pol III. Additionally, while the study did include ChIP-E2F1 data, it did not provide PCR primer information, which makes it difficult to compare the ChIP footprint with the predicted binding sites.

### 2.5. Factors to Determine nc886 Expression; CpG DNA Methylation

There is a CpG island at the nc886 promoter region (nt coordinates from −189 to +82, with the 5′-end of nc886 RNA being +1; Figure 1A). CpG hypermethylation suppresses nc886 transcription. The most direct evidence is probably the restoration of nc886 expression when DNA methylation was inhibited by treatment of various cell lines with 5-azacytidine or 5-aza-2′-deoxycytidine (AzaC or AzadC). They included HL60 acute myeloid leukemia (AML) cells [21], NCI-H446 lung cancer cells [34], TT and TE-8 esophageal cancer cells [41], SNU-005 and SNU-484 gastric cancer cells [40] HuT-78 CD4+ T-cells [64], 184FMY2 and 184B5 HMECs [20], LNCaP prostate cancer cells [48], PC-3M-1E8 and PC-3M-2B4 (2B4) prostate cancer cells [86], and SKOV3 and A2780 ovarian cancer cells [7].

In addition, a negative correlation between nc886 RNA expression and the promoter CpG methylation has been observed in cancer cell lines and patient specimens of various tissue origins, including blood [21], lung [34], esophagus [41], stomach [40], breast [20], and prostate tissues [48,52,86]. This inverse correlation was also shown in blood samples of several cohorts, including the Young Finns study (YFS) cohort (*n* = 47) [67], women who experienced pre-term or term birth (*n* = 20 each) [66], type 2 diabetes mellitus (T2DM) patients who received GLP1 analog therapy (*n* = 138) [87], and IgA nephropathy (IgAN) patients or healthy subjects (*n* = 10 each) [64].

In a number of cell culture studies (as listed in Table 3), the alteration of nc886 expression was examined, and some of these studies also measured changes in nc886 DNA methylation and found that they are inversely correlated (as shown in Table 3). One notable example is the induction of nc886 by transforming growth factor-β (TGF-β). The TGF-β idea came from the genomic location of nc886 between TGFBI (a TGF-β-induced gene) and SMAD5 (Figure 1A), both of which are implicated in the TGF-β signaling [11]. TGF-β treatment leads to demethylation of nc886 in ovarian cancer cells [7]. The expression levels of nc886 and TGFBI were elevated together by TGF-β in an ovarian cancer cell line and were positively associated in ovarian cancer patient samples. Together with the fact that TGFBI is also regulated by its promoter CpG methylation [88], all these data suggest that nc886 and TGFBI are co-regulated by TGF-β via an epigenetic mechanism. Importantly, the increase in nc886 expression by TGF-β was negated upon ectopic expression of a CpG methyltransferase DNMT1, providing concrete evidence for causal relationship between DNA methylation and RNA expression [7]. The cell culture data on TGF-β and nc886 were complemented by a recent cohort study in which nc886 RNA levels in peripheral blood samples of T2DM patients (*n* = 138) were positively correlated with serum TGF-β1 levels measured by enzyme-linked immunosorbent assay [87].

Besides the literature we have discussed above, there are a number of human cohort studies with nc886 CpG methylation data. A majority of these studies are epigenome-wide screenings that identified nc886 to be one of the most differentially methylated regions (DMRs). We will not further discuss them here (this will be elaborated in another future review) because the topic of this review is nc886’s variable expression, and most of them did not measure nc886 RNA. Nevertheless, we are confident that nc886 RNA levels will be accordingly different among subject individuals in those epigenome-wide studies because the suppression of nc886 expression by CpG methylation is undoubted, as we discussed above. We surmise that CpG methylation is probably the most critical determinant for an individual’s basal nc886 expression level in non-pathological conditions.

### 2.6. Factors to Determine nc886 Expression; RNA Stability

nc886 RNA is very unstable. An nc886 measurement upon transcription shutdown by actinomycin D estimated its half-life to be approximately one hour in several cell lines [5,20]. This observation was corroborated by doxorubicin treatment that lead a to >50% decrease of nc886 within two hours [6]. This instability is in sharp contrast to classic Pol III-transcribed ncRNAs (tRNAs, 5S rRNA, snRNAs, etc.) that are highly stable. As shown in the case of doxorubicin, nc886’s short half-life renders its cellular level to change instantly in response to external stimuli.

nc886’s instability is most probably due to degradation by nuclease(s). Regarding putative nucleases, the pattern of small RNA-seq reads provides a clue (Figure 1D). Seq reads mapped to miR-886-5p start from the almost invariant 5′-nt (which is nc886’s 5′-end) but terminate variably at the 3′-side. This pattern mirrors at the other side; seq reads to miR-886-3p start variably at the 5′-side but terminate constantly at nc886’s 3′-end. This indicates that both ends of nc886 are protected by a certain mechanism. This also suggests initial endonucleolytic cleavage followed by processive exonucleolytic digestion from the cleavage site.

So far, the only nuclease with experimental evidence is Dicer (Figure 1D), which degraded nc886 RNA when mixed in vitro [5,7]. Again, we want to highlight that nc886 degradation does not yield miR-886-5p or -3p and is clearly distinct from Dicer’s processing of a canonical miRNA precursor. In addition, nc886 levels increased and decreased respectively upon the knockdown and ectopic expression of Dicer in ovarian cell lines [7]. However, the fold-change of nc886 upon Dicer knockdown or expression was modest. Most probably, there are other nucleases targeting nc886. Although nc886 was unstable in all the cell lines examined so far [5,6,20], we will not rule out a biological situation in which the activity of nc886-targeting nucleases is low and, resultantly, nc886 becomes stable.

### 2.7. Factors to Determine nc886 Expression; Genetic Variation

As described earlier, miR-886-5p and -3p were examined in the YFS cohort (*n* = 765) [67]. This research group performed a genome-wide association study (GWAS), leading to the identification of 130 and 180 overlapping genetic variations that are associated with the expression levels of miR-886-5p and -3p, respectively. The majority of the 180 and 130 variants, especially those with strong association, were located within a 100 kb region from 92 to 193 kb downstream of the nc886 locus. Although the association of these variations was statistically significant, it remains to be validated whether they influence nc886 expression levels to a quantitatively meaningful extent.

## 3. Conclusions and Future Perspectives

The dynamic expression of nc886 RNA is absolutely unique among Pol III genes, as discussed in this review. Its transcription is controlled by TFs, CpG DNA methylation, and other factors. Additionally, the steady-state level of functional nc886 in a cell is determined by its degradation rate and subcellular localization. nc886 is not only diffused in the cytoplasm but is also concentrated in cytoplasmic foci [5]. Collectively, it would not be an exaggeration to state that the nc886 RNA levels at a given moment in each cell are all different.

Our current knowledge about nc886 regulation is primitive. Elucidating it would be a challenging task that requires far more research endeavors. There are numerous questions to consider, but we have highlighted a few that we believe are important: Which TF controls nc886 expression in diverse biological contexts? Which nuclease degrades nc886? Regarding regulation by CpG methylation, there is a shortage of basic studies on methylation enzymes, upstream signals, and so on.

In regard to studying nc886 regulation, there are practical hurdles and to-be-considered points. When assessing nc886 RNA levels, especially in clinical samples, caution is needed because nc886 is highly unstable and prone to degradation. nc886 degradation during sample preparation and RNA measurement will cause misleading inaccuracies when analyzing nc886 methylation and TF activity. Another obstacle is that there is no conservation of the nc886 gene in mice [13,89]. Thus, animal studies are limited, and in vivo evidence relies heavily on data from human cohorts and clinical specimens. RNA-seq from human samples could have been a rich source for nc886 expression data. However, nc886 reads are captured far lower than its actual abundance because nc886 is a medium-sized ncRNA without a polyA tail. Based on our experience (YSL, unpublished data), it is not entirely reliable to assess nc886 expression levels from current RNA-seq data. In the past, miRNA array platforms with miR-886-5p and -3p were quite useful, but they are no longer available regrettably. The development and widespread use of array or RNA-seq platforms specifically for the Pol III transcriptome could greatly expand our understanding of nc886 regulation and that of other Pol III genes.

## Figures and Tables

**Figure 1 ijms-24-08533-f001:**
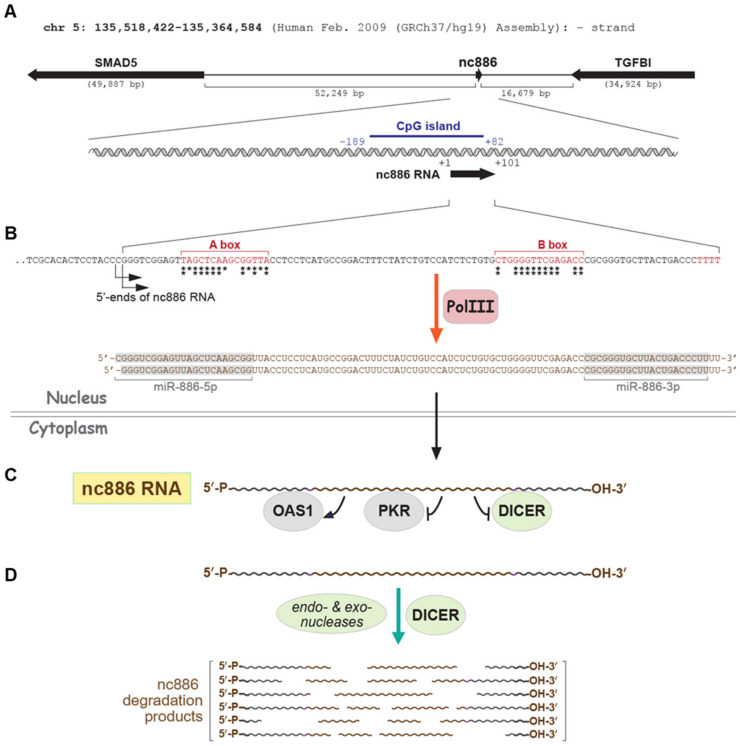
Overview of nc886 expression and function. (**A**) nc886’s genomic location and CpG island. The arrows for nc886, TGFBI, and SMAD5 designate transcription directions. (**B**) nc886’s sequence in black for DNA and in brown for RNA. Red letters, cis-acting promoter elements and the transcription termination signal; two asterisks and one asterisk below A and B box, the best and the second-best consensus nts, respectively. (**C**) nc886’s interacting proteins. Arrow, activator; blunt-ended lines, suppressors. Innate immunity proteins in grey and nucleases in light green. (**D**) Decay of nc886.

**Figure 2 ijms-24-08533-f002:**
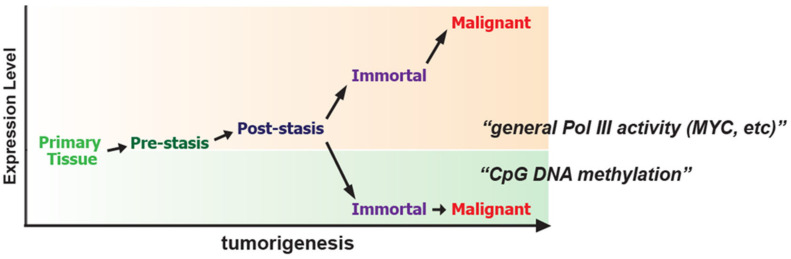
Two opposite directions of nc886 expression during tumorigenesis.

**Figure 3 ijms-24-08533-f003:**
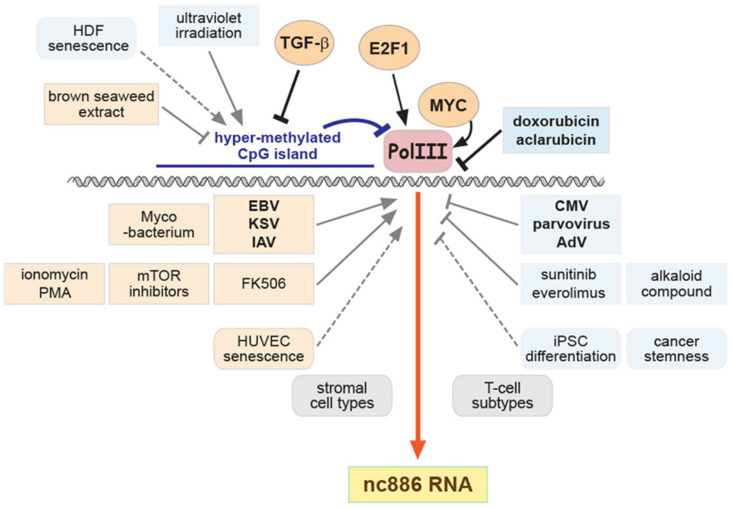
A cartoon summarizing factors that affect nc886 expression. nc886 expression is up- and downregulated (arrows and blunt-ended lines, respectively) by various factors (solid lines) or in cellular conditions (dotted lines). nc886-activating and -suppressive situations are illustrated in orange and blue tones, respectively. Mixed situations are in grey.

**Table 1 ijms-24-08533-t001:** nc886 expression in clinical specimens from cancer patients. (T, tumor samples; *n*, normal samples; overall survival, OS; progression-free survival, PFS; recurrence-free survival, RFS).

Disease	Specimens	Method to Measure	Expression	Comments	Reference
squamous cell lung carcinoma	paired T vs. N (*n* = 4)	miRNA profiling	miR-886-5p and -3p down in T		[27]
cervical squamous cell carcinoma	paired T vs. N (*n* = 12)	miRNA profiling and qRT-PCR	miR-886-5p up in T		[28]
prostate cancer	paired T vs. N (*n* = 3)	Northern for nc886	nc886 down in T		[5]
papillary thyroid cancer	T (*n* = 28) vs. N (*n* = 10)	miRNA profiling and qRT-PCR	miR-886-3p down in T	down in sporadic type (*n* = 21)	[29]
hepatocellular carcinoma recurrence (RC)	RC (*n* = 52) vs. non-RC (*n* = 63)	miRNA profiling and qRT-PCR	miR-886-5p down in recurrence	RC following orthotopic liver transplantation	[30]
oral leukoplakia (OLK)	malignantly transformed (mt) OLK (*n* = 7) vs. OLK (*n* = 20)	miRNA profiling	miR-886-5p and -3p up in mtOLK		[31]
cholangio-carcinoma	paired T vs. N (*n* = 16)	nc886 qRT-PCR	nc886 variably expressed		[32]
bladder cancer with cisplatin therapy	progression of disease (*n* = 7) vs. complete responses (*n* = 8)	miRNA profiling	miR-886-5p and -3p down in complete responses		[33]
small cell lung cancer (SCLC)	SCLC patient FFPE (*n* = 82)	miRNA profiling and qRT-PCR	miR-886-3p variably expressed	low miR-886-3p = short OS and PFS	[34,35]
anaplastic large cell lymphoma	anaplastic lymphoma kinase (ALK)-positive (*n* = 33) vs. -negative (*n* = 25)	miRNA profiling	miR-886-5p and -3p up in ALK-positive		[36]
breast cancer	normal (*n* = 29) vs. benign (*n* = 21) vs. malignant (*n* = 29)	miRNA profiling	miR-886-3p up (normal → benign) and down (benign → malignant)		[37]
thyroid cancer	poorly-differentiated (pd) (*n* = 14) vs. oncocytic pd (*n* = 13) vs. well-differentiated (*n* = 72) vs. normal (*n* = 8)	miRNA profiling	miR-886-3p down in pd		[38]
colorectal cancer (CRC)	Metastatic CRC (*n* = 138)	miRNA profiling	miR-886-3p variably expressed	high miR-886-3p (in blood) = short OS	[39]
gastric cancer	paired T vs. N (*n* = 88)	nc886 qRT-PCR	nc886 down in T		[40]
esophageal squamous cell carcinoma	paired T vs. N (*n* = 84)	nc886 Northern	nc886 down in T	low nc886 = short RFS	[41]
bladder tumor	patient FFPE (*n* = 70)	miRNA qRT-PCR	miR-886-5p variably expressed	miR-886-5p up in high-grade tumors	[42]
cervical cancer	paired T vs. N (*n* = 31)	in situ hybridization	miR-886-5p up in T		[43]
esophageal squamous cell carcinoma	discovery set (*n* = 108) validation set (*n* = 214)	nc886 qRT-PCR	nc886 variably expressed	nc886 as a predictor for RFS	[44]
esophageal squamous cell carcinoma	post-operative: non-relapse (*n* = 5) vs. relapse (*n* = 14)	miRNA profiling	miR-886-3p down in non-relapse		[45]
renal cell carcinoma	paired T vs. N (*n* = 36)	miRNA qRT-PCR	miR-886-3p up in T		[46]
thyroid cancer	paired T vs. N (*n* = 37)	nc886 qRT-PCR	nc886 variably expressed	high nc886 = advanced T staging and metastasis	[12]
renal cell carcinoma	paired T vs. N (*n* = 20); stage I/II (*n* = 11) vs. stage III/IV (*n* = 9)	nc886 qRT-PCR	nc886 level: stage III/IV > stage I/II > *n*		[47]
prostate cancer	paired T vs. N (*n* = 6);	nc886 qRT-PCR	nc886 down in T		[48]
ovarian cancer	T (*n* = 25)	nc886 qRT-PCR	nc886 variably expressed	high nc886 = poor prognosis	[7]
prostate cancer	low-grade (*n* = 40) vs. high-grade (*n* = 26)	miRNA profiling	miR-886-3p up in high-grade	measured in plasma	[49]
multiple myeloma (MM)	blood of MM patients (*n* = 10) vs. healthy donors (*n* = 6)	miRNA qRT-PCR	miR-886-5p up in MM patients		[50]
multiple myeloma (MM)	Plasma cells from MM patients (*n* = 153)	miRNA profiling	miR-886-5p and -3p variably expressed	high miR-886-5p and -3p = short OS	
colorectal cancer (CRC)	blood, T, N from CRC patients (*n* = 197)	nc886 qRT-PCR	nc886 variably expressed	high nc886 = good response to 5-fluorouracil	[51]
prostate cancer	paired T vs. N (*n* = 6)	miRNA qRT-PCR	miR-886-3p down in T		[52]
oral squamous cell carcinoma	paired T vs. N (*n* = 54)	miRNA qRT-PCR	miR-886-3p down in T		[53]
prostate cancer	normal (*n* = 29) vs. primary (*n* = 131) vs. metastatic (*n* = 19) prostate cancer	miRNA profiling	miR-886-3p: normal > primary > metastatic		[54]
prolactin pituitary tumor	aggressive (*n* = 4) vs. non-aggressive (*n* = 8) tumors	miRNA profiling	miR-886-3p up in aggressive tumors		[55]

**Table 2 ijms-24-08533-t002:** nc886 expression in disorders.

Disease	Specimens	Method to Measure	Expression	Comments	Reference
acute cellular rejection (ACR)	ACR (*n* = 15) vs. non-ARC (*n* = 11)	miRNA profiling and qRT-PCR	miR-886-5p and -3p up in ARC	FFPE intestinal mucosal biopsy	[56]
allergic rhinitis (AR)	AR (*n* = 10) vs. non-AR (*n* = 10)	miRNA profiling	miR-886-3p down in AR	nasal mucosa	[57]
Friedreich ataxia (FRDA)	FRDA (*n* = 8) vs. normal (*n* = 4)	miRNA profiling and qRT-PCR	miR-886-5p and -3p up in FRDA	lymphoblasts or PBL	[58]
Parkinson disease	Parkinson disease patients (*n* = 11) vs. normal (*n* = 6)	miRNA profiling and qRT-PCR	miR-886-5p up in patients	amygdala from patients at motor stages of the disease	[59]
asthma and allergic rhinitis (AR)	AR (*n* = 117) vs. normal (*n* = 33)	miRNA profiling	miR-886-3p down in AR	men aged 37–48 yrs, with ~20 yrs of asthma	[60]
myotonic dystrophy type 1 (DM1)	DM1 patients (*n* = 36) vs. age/sex matched control (*n* = 36)	miRNA profiling	miR-886-3p up in DM1	plasma	[61]
rheumatoid arthritis (RA)	RA patients treated with adalimumab (*n* = 89) vs. placebo (*n* = 91)	miRNA profiling	miR-886-3p variably expressed	high miR-886-3p (+low miR-22) = good response	[62]
neuro-behavioral development	infants (*n* = 615); nc886 measured in placenta	Custom Nanostring (144 imprinted genes)	nc886 variably expressed	nc886 as a factor cluster behavioral profiles	[63]
IgA nephropathy (IgAN)	IgAN patients (*n* = 10) vs. healthy subjects (*n* = 10)	miR-886 precursor qRT-PCR	miR-886 precursor down in IgAN	Measured in CD4+ T-cells	[64]
acute kidney injury (AKI) upon sepsis	AKI patients (*n* = 235) vs. non-AKI (*n* = 235)	miRNA qRT-PCR	miR-886-5p up in non-AKI	measured in plasma	[65]
Preterm birth (PTB)	women with PTB (*n* = 20) vs. term delivery (*n* = 20)	nc886 qRT-PCR	nc886 down in PTB	measured in blood	[66]

**Table 3 ijms-24-08533-t003:** nc886 expression according to biological and environmental factors.

Trigger or Factor	System	Method to Measure	Expression	Comments	Reference
differentiation	from induced pluripotent stem cells (iPSC) to hepatocytes	miRNA qRT-PCR? (not clear)	miR-886-5p and -3p down in hepatocytes		[67]
cancer stem cells	colon cancer cell line SW1116 vs. its derivative cancer stem cells (SW1116csc),	miRNA profiling	miR-886-5p and -3p down in SW1116csc		[68]
bone marrow stromal cells types	HS27a cells (CXCL12-positive) vs. HS5 cells (-negative)	miRNA profiling and nc886 Northern	nc886 up in CXCL12-negative cells	in quantity, nc886 much higher than miR-886-3p	[69]
Stimulation of CD4+ T cells	stimulated by ionomycin and phorbol myristate acetate	nc886 qRT-PCR and Northern	nc886 up in stimulated T cells		[16]
T cell subtypes interleukin-22 (IL22) (+) or (−)	[CD3(+)IL22(+)] vs. [CD3(+)IL22(−)] T-cell populations	miRNA profiling	miR-886-3p and -5p up in IL22(+) cells.		[70]
cellular senescence	multiple passages of primary human umbilical vein endothelial cells until replicative senescence	miRNA profiling	miR-886-5p up in senescence		[71]
cellular senescence	multiple passages of human dermal fibroblasts until replicative senescence	nc886 qRT-PCR	nc886 down in senescence	nc886 DNA CpG methylation up	[72]
transforming growth factor-β (TGF-β)	TGF-β treatment in SKOV3 ovarian cancer cells	nc886 Northern	nc886 up upon treatment	nc886 DNA CpG methylation down	[7]
DNA-reactive compounds	doxorubicin and aclarubicin treatment in various cells	nc886 Northern	nc886 down upon treatment	eviction of Pol III from DNA	[6]
alkaloid compound arecoline (ARE)	ARE treatment in oral squamous cell carcinoma cell lines	miRNA qRT-PCR	miR-886-3p down upon treatment		[53]
ultraviolet (UV) radiation	UV irradiation on HaCaT keratinocyte cells	nc886 qRT-PCR	nc886 down by UV	nc886 DNA CpG methylation up	[73,74]
cancer therapy drugs	Sunitinib and Everolimus treatment in renal cancer cells	nc886 qRT-PCR	nc886 down upon treatment		[75]
immune-suppressants	FK506 and mammalian target of rapamycin (mTOR) inhibitors treatment in HepG2 cells	miRNA profiling	miR-886-5p and -3p up upon treatment		[76]
areca nut extracts (ANE)	ANE treatment in normal human gingival fibroblasts	miRNA profiling	miR-886-3p down upon treatment		[77]
brown seaweed (*Laminaria japonica*) extract	*Laminaria japonica* extract treatment in HaCaT keratinocyte cells	nc886 qRT-PCR	nc886 up upon treatment	nc886 DNA CpG methylation down	[74]
γ-herpesviridae family viruses	EBV and Kaposi’s sarcoma virus infection into lymphocytes and B cell lines.	nc886 Northern	nc886 up upon infection	other vtRNAs also up	[78]
cytomegalo-virus (CMV)	CMV infection into primary human foreskin fibroblast cells	nc886 Northern	nc886 down upon infection	other vtRNAs also down	[78]
parvovirus	parvovirus infection into human newborn kidney cells	nc886 Northern	nc886 down upon infection	other vtRNAs unaffected	[78]
*Mycobacterium avium hominissuis*	mycobacterial infection into human monocyte-derived macrophages	miRNA profiling and qRT-PCR	miR-886-5p up upon infection		[79]
influenza A virus (IAV)	IAV infection into several human cell lines	nc886 qRT-PCR and Northern	nc886 up upon infection	other vtRNAs also up	[14]
adenovirus (AdV)	AdV infection into thyroid cell lines	nc886 Northern	nc886 down upon infection		[13]

## Data Availability

Not applicable.

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
