# Peer review of "nc886, an RNA Polymerase III-Transcribed Noncoding RNA Whose Expression Is Dynamic and Regulated by Intriguing Mechanisms"

_ijms, 2023, doi:10.3390/ijms24108533_

Round 1

Reviewer 1 Report

This review investigates the function of nc886, an RNA polymerase III-transcribednoncoding RNA in  physiological  and pathological  conditions. They reviewed the possible functions of nc886 in multiple mechanisms including as promoter  CpG  DNA  methylation, transcription factor activity and RNA instability. Authors propose that nc886 have therapeutic promise in regulating/overcoming tumorigenesis, innate immunity and may give a novel approach to tailored treatment strategies. Although this is a very important topic, some issues still exist and need to be carefully discussed and revised.   

Major issues:  

1. A schematic diagram needs to be added to comprehensively explain the molecular mechanisms.

2. The paper must be written in standard English.

Author Response

Dear Editor,

Thank you for considering our manuscript for publication. I also thank reviewers for their helpful comments. Below are our responses to reviewers. For easy reading, our point-by-point responses are in brackets highlighted by four asterisks ****[….] to distinguish them from the original comments. 

Reviewer #1:

Comments and Suggestions for Authors

This review investigates the function of nc886, an RNA polymerase III-transcribed noncoding RNA in physiological and pathological conditions. They reviewed the possible functions of nc886 in multiple mechanisms including as promoter CpG DNA methylation, transcription factor activity and RNA instability. Authors propose that nc886 have therapeutic promise in regulating/overcoming tumorigenesis, innate immunity and may give a novel approach to tailored treatment strategies. Although this is a very important topic, some issues still exist and need to be carefully discussed and revised.  

Major issues: 

  1. A schematic diagram needs to be added to comprehensively explain the molecular mechanisms.

**** [During the initial submission, we had included figure files. However, for some reason, they disappeared from the manuscript file for review. We have included figures (page 14~16 in the revised manuscript).]

  1. The paper must be written in standard English.

**** [We had this paper through extensive English editing (see track changes).]

Reviewer 2 Report

1.     No figure is attached.

2.     The authors mentioned in the manuscript that the level of nc886 is highly dynamic within cells. I’m concerned whether the degradation product of nc886 will result in the inaccuracy of using miR886-5p and miR886-3p expression as a surrogate for nc886’s expression.

3.     What is the exact mechanism of nc886 mediated PKR and downstream signaling? Just by binding and sequestering PKR?

Author Response

Dear Editor,

Thank you for considering our manuscript for publication. I also thank reviewers for their helpful comments. Below are our responses to reviewers. For easy reading, our point-by-point responses are in brackets highlighted by four asterisks ****[….] to distinguish them from the original comments. 

Reviewer #2:

Comments and Suggestions for Authors

  1. No figure is attached.

**** [During the initial submission, we had included figure files. However, for some reason, they disappeared from the manuscript file for review. We have included figures (page 14~16 in the revised manuscript).]

  1. The authors mentioned in the manuscript that the level of nc886 is highly dynamic within cells. I’m concerned whether the degradation product of nc886 will result in the inaccuracy of using miR886-5p and miR886-3p expression as a surrogate for nc886’s expression.

**** [We are confident that those nc886 degradation products are negligible in quantity, as compared to the intact nc886, based on Northern data from multiple cell lines. We mentioned this in the revised text: line 137~140 “Our Northern blot experiments never observed smaller products than nc886 at 101 nt size, and therefore we are confident that it was intact nc886, but neither miR-886-5p/-3p nor degradation products, which was detected in most miRNA array experiments [5,7,26].”]

  1. What is the exact mechanism of nc886 mediated PKR and downstream signaling? Just by binding and sequestering PKR?

**** [nc886 binds directly to PKR and prevents it from being activated. However, this review focuses on “the regulation of nc886 expression”, rather than nc886’s function. We are planning another future review paper focusing on nc886’s molecular function and cellular roles]